# Assessing Drug Interaction and Pharmacokinetics of Loxoprofen in Mice Treated with CYP3A Modulators

**DOI:** 10.3390/pharmaceutics11090479

**Published:** 2019-09-16

**Authors:** Sanjita Paudel, Aarajana Shrestha, Piljoung Cho, Riya Shrestha, Younah Kim, Taeho Lee, Ju-Hyun Kim, Tae Cheon Jeong, Eung-Seok Lee, Sangkyu Lee

**Affiliations:** 1BK21 Plus KNU Multi-Omics based Creative Drug Research Team, College of Pharmacy, Research Institute of Pharmaceutical Sciences, Kyungpook National University, Daegu 41566, Korea; sanjitapdl99@gmail.com (S.P.); whvlfwjd@naver.com (P.C.); riya.shrestha07@gmail.com (R.S.); younah86@naver.com (Y.K.); tlee@knu.ac.kr (T.L.); 2College of Pharmacy, Yeungnam University, Gyeongsan 38541, Korea; aarajanashrestha1@gmail.com (A.S.); jhkim@yu.ac.kr (J.-H.K.); taecheon@ynu.ac.kr (T.C.J.); eslee@ynu.ac.kr (E.-S.L.)

**Keywords:** Loxoprofen, drug-drug interaction, CYP3A, Dexamethasone, Ketoconazole

## Abstract

Loxoprofen (LOX) is a non-selective cyclooxygenase inhibitor that is widely used for the treatment of pain and inflammation caused by chronic and transitory conditions. Its alcoholic metabolites are formed by carbonyl reductase (CR) and they consist of trans-LOX, which is active, and cis-LOX, which is inactive. In addition, LOX can also be converted into an inactive hydroxylated metabolite (OH-LOXs) by cytochrome P450 (CYP). In a previous study, we reported that CYP3A4 is primarily responsible for the formation of OH-LOX in human liver microsomes. Although metabolism by CYP3A4 does not produce active metabolites, it can affect the conversion of LOX into trans-/cis-LOX, since CYP3A4 activity modulates the substrate LOX concentration. Although the pharmacokinetics (PK) and metabolism of LOX have been well defined, its CYP-related interactions have not been fully characterized. Therefore, we investigated the metabolism of LOX after pretreatment with dexamethasone (DEX) and ketoconazole (KTC), which induce and inhibit the activities of CYP3A, respectively. We monitored their effects on the PK parameters of LOX, cis-LOX, and trans-LOX in mice, and demonstrated that their PK parameters significantly changed in the presence of DEX or KTC pretreatment. Specifically, DEX significantly decreased the concentration of the LOX active metabolite formed by CR, which corresponded to an increased concentration of OH-LOX formed by CYP3A4. The opposite result occurred with KTC (a CYP3A inhibitor) pretreatment. Thus, we conclude that concomitant use of LOX with CYP3A modulators may lead to drug–drug interactions and result in minor to severe toxicity even though there is no direct change in the metabolic pathway that forms the LOX active metabolite.

## 1. Introduction

Loxoprofen (LOX) is an anti-inflammatory prodrug (NSAID) with potential antipyretic and analgesic properties, but its side effects are less defined when compared to other NSAIDs [1,2,3]. It is one of the most clinically prescribed NSAIDs in Japan, and it is also popular in Eastern Asia, the Middle East, Latin America, and Africa [1]. It is a well-known cyclo-oxygenase (COX1 and COX2) inhibitor that reduces the synthesis of inflammatory agents such as prostaglandins [2,4,5]. It is also widely used for the management of pain and inflammation in chronic and transient conditions (e.g., toothache, headache, menstrual cramps, common cold, etc.) [1,2,6].

Although LOX has fewer sides effects than other NSAIDs, many adverse effects have been reported such as GI disorders (erosive gastritis and bleeding), renal disorders, cardiovascular disorders, headaches, anaphylaxis, and abdominal pain [1,2,6,7]. Moreover, the use of LOX with drugs such as methotrexate, warfarin, aspirin, and valacyclovir is contraindicated [6,8,9,10]. Generally, the pharmacokinetic (PK) parameters of many drugs are influenced by the inhibition or induction of cytochrome P450 enzymes (CYPs) [11], which can result in various drug interactions, toxicity, and either an increase or decrease in drug activity [12].

Previous studies have extensively investigated the metabolic activities of LOX in various organisms such as rats, mice, monkeys, rabbits, and humans [1,13,14]. However, to the best of our knowledge, there is still a large gap in knowledge regarding the interactions of LOX with other drugs that may cause simple to severe adverse drug interactions. The alcoholic metabolites of LOX (cis-LOX and trans-LOX) are mainly produced by carbonyl reductase (CR), and trans-LOX is the only active metabolite derived from LOX, which LOX is a pharmacologically inactive drug unless it is metabolized to trans-LOX [1,5]. In addition to CR, LOX is also metabolized by CYP450. In our previous study, we reported that the CYP-mediated metabolism of LOX was catalyzed by CYP3A4 and CYP3A5 to form hydroxylated LOX (OH-LOX). We also found that an inactive metabolite of LOX catalyzed by CYP was significantly higher in dexamethasone (DEX)-induced liver microsomes [15].

It is well known that CYP3A4 is most abundant in the liver, and it can catalyze approximately 50% of commercially available drugs [16,17]. Many commercial drugs are known to regulate the activity of CYP3A, and they may be clinically administered with LOX. The CYP3A/LOX metabolic pathway produces an inactive metabolite and competes with another pathway for LOX that produces the active metabolite. If LOX is co-administered with a drug that modulates the activity of CYP3A, there may be possible drug interactions affecting the concentration of LOX or the active metabolite of LOX. Thus, we investigated the interaction of LOX with CYP3A by monitoring the effects of DEX (a CYP3A inducer) or ketoconazole (KTC, a CYP3A inhibitor) pretreatment on the PK parameters of LOX, cis-LOX, and trans-LOX in an ICR mouse model.

## 2. Materials and Methods

### 2.1. Materials

LOX (2-(4-((2-Oxocyclopentyl)methyl)phenyl)propanoic acid, C_15_H_18_O_3_, CAS ID 68767-14-5) was procured from Tokyo Chemicals Industry (Tokyo, Japan). LOX was used to synthesize cis-LOX and trans-LOX ((2*S*)-2-[4-[[(1*R*,2*S*)-2-hydroxycyclopentyl]methyl]phenyl]propanoic acid, C_15_H_20_O_3_, CAS ID 83648-76-4) with purities of 96.5% and 97.9%, respectively [18]. DEX, KTC, dextromethorphan, and phenacetin were purchased from Sigma-Aldrich Co., LLC. (St. Louis, MO, USA) while midazolam was procured from Bukwang Pharmaceutical Co., Ltd. (Seoul, Korea). Mass spectrometry (MS) grade water and acetonitrile (ACN) were obtained from Fischer Scientific (Pittsburgh, PA, USA).

### 2.2. Animal Treatments and Sample Preparations

Male ICR mice (36 mice of 5 weeks) were purchased from Orient Co. (Seongnam, Korea) andwere randomly divided into 4–6 mice per cage. Then, the mice were acclimatized for 1 week in a controlled environment (relative humidity: 60%, temperature: 25 °C) under a 12-h/12-h light/dark cycle and supplied standard rodent chow and tap water freely. All animal handling procedures followed protocols issued by the Society of Toxicology (USA, 1989) and were approved on March 21, 2019 by the Institutional Review Board of Kyungpook National University (project ID # 2019-41).

The acclimatized animals were divided into eight groups, with each group containing 3 mice: groups I and II, vehicle (corn oil); groups III and IV, DEX-treated; groups V and VI, vehicle (10% Ethanol); and groups VII and VIII, KTC-treated group. Briefly, groups I and II were intraperitoneally (*i.p*.) treated with corn oil for 3 days while DEX (dissolved in corn oil and administered at 40 mg/kg) was given to groups III and IV for 3 consecutive days [19]. Then, animals were fasted for 12 hours with free access to water before starting the experiment. After fasting, LOX (20 mg/kg) was administered orally to groups I, II, III, and IV. Ethanol (10%) and KTC (60 mg/kg) were administered *i.p.* to the vehicle-treated groups (V and VI) and KTC-treated groups (VII and VIII), respectively [20], and after 3 min, LOX was given orally (20 mg/kg).

After the oral administration of LOX, blood from groups I, III, V, and VII was collected from the tail at 0, 5, 10, 15, 30, 60, 120, and 240 min post-administration and placed into sodium heparin-containing tubes. After the last blood collection, the mice were sacrificed by cervical dislocation. Plasma was then prepared by centrifuging the blood at 4000× *g* for 15 min at 4 °C and stored at −80 °C until analysis. For analysis, each sample was prepared by mixing plasma (10 µL) and 90 µL of ACN containing 0.1% formic acid and 5 µM of tolbutamide as an internal standard (IS). The samples were then vortexed and centrifuged at 13,000× *g* for 10 min at 4 °C. Finally, 10 µL of sample supernatant was injected into the LC-MS/MS system. The PK parameters (the maximum plasma concentration [*C*_max_], time to reach the maximum plasma concentration [*T*_max_], elimination half-life [*T*_1/2_], and area under the plasma concentrations [AUC]) were analyzed by WinNonlin software (Version 2.1, Scientific Consulting, Louisville, KY, USA).

Similarly, blood from groups II, IV, VI, and VIII was collected from the hepatic portal vein 10 min post-LOX administration to identify metabolites of LOX and to analyze drug–drug interactions. Plasma from these samples was prepared as explained above. Then, 300 µL of ACN having 0.1% formic acid and 5 µM of the IS were mixed with 100 µL of each plasma sample. Next, the samples were vortexed and centrifuged at 13,000× *g* for 10 min at 4 °C. Supernatants were transferred into tubes and dried using a Labconco Speed Vac (Labconco Corporation, Kansas City, MO, USA). The dried samples were reconstituted using 100 µL of 50% methanol and centrifuged at 13,000× *g* for 10 min at 4 °C. Each supernatant was transferred into an LCMS vial, and 5 µL were injected into a high-resolution mass spectrometer (HRMS).

### 2.3. CYP Activities in the Mouse Liver

Male ICR mice were divided into 4 groups with each group containing 3 mice: group I, vehicle (corn oil); group II, DEX-treated; group III, vehicle (10% Ethanol); and group IV, KTC-treated group. Group I was treated with corn oil for 3 days while DEX (dissolved in corn oil and administered at 40 mg/kg) was administered *i.p.* to group II for 3 consecutive days [19]. Similarly, group III and group IV were treated once with 10% ethanol or KTC (60 mg/kg), respectively [20]. Twenty-four hours after the last treatment, the liver was excised and homogenized with three volumes of ice-cold 0.1 M potassium phosphate buffer (pH 7.4). The supernatant fraction was then separated as the S9 fraction from the mixture by centrifugation at 9000× *g* at 4 °C and stored at −80 °C until use.

To characterize the CYP activities in the harvested livers, phenacetin *O*-demethylation (using 80 µM of phenacetin) for CYP1A, dextromethorphan *O*-demethylation (using 5 µM of dextromethorphan) for CYP2D, and midazolam 1’-hydroxylation (using 5 µM of midazolam) for CYP3A were used as cocktail probe reactions [21,22]. The level of protein in the S9 fraction was determined by Bradford assay [23]. The cocktail probes were then incubated with 10 µL of each S9 fraction, potassium phosphate buffer (pH 7.4), and an NADPH generating system (NGS) in a final volume of 100 µL. After incubation for 30 min at 37 °C, ice-cold ACN containing 0.1% formic acid and IS (5 µM) was added to stop the reaction. Samples were then vortexed and centrifuged at 13,000× *g* for 10 min at 4 °C. Finally, the supernatants were transferred to LC-MS/MS vials for analysis.

### 2.4. Instrument and Data Acquisition

A Shimadzu Prominence UFLC system (Kyoto, Japan) connected to a TSQ vantage triple quadrupole mass spectrometer with a HESI-II spray source incorporated with a DGU-20A_5_ degasser, an LC-20AD pump, a SIL-20A autosampler, and a CTO-20A column oven was used for the analyses. A shim-pack GIS C18 column (150 × 3.0 mm, 3 µM) was used to separate analytes in the samples. Mobile phases A and B were composed of water with formic acid and ACN with 0.1% formic acid, respectively, and the flow rate was 0.50 mL/min at 40 °C. The gradient conditions were as follows: 20% of B between 0 and 0.25 min, 20–80% of B between 0.25 and 9.75 min, 80–20% of B between 9.75 and 10 min, and 20% of B between 10 and 13 min. The MS was operated under the following conditions: electrospray ionization in negative mode at 3.0 kV, capillary temperature at 350 °C, vaporizer temperature at 300 °C, sheath gas pressure at 35 Arb, and auxiliary gas pressure at 10 Arb. Finally, Xcalibur software (Thermo Fisher Scientific Inc., Waltham, MA, USA) was used for data analysis.

HRMS coupled with ultrahigh performance liquid chromatography (UHPLC) was used to detect hydroxy-LOX and other metabolites of LOX. The UHPLC system, Dionex Ultimate 3000 (Dionex Softron GmbH, Germering, Germany) consisted of an HPG-3200SD Standard binary pump, a WPS 3000 TRS analytical autosampler, and a TCC-3000 SD column compartment. In this experiment, the HRMS was a Q Exactive Focus quadrupole-Orbitrap MS (Thermo Fisher Scientific, Bremen, Germany) equipped with a heated electrospray ionization (HESI-II) ion probe.

LOX was detected as a deprotonated ion [M-H]^−^ at *m*/*z* 245.1175 in negative ion mode. Therefore, negative ion mode was used with the following optimized conditions for LOX: spray voltage of 2.5 kV, capillary temperature of 320 °C, auxiliary gas at 12 aux unit, aux gas heater temperature of 200 °C, sheath gas at 35 aux units, and S-lens RF level of 50. LOX and its metabolites were separated using a 150 mm × 2.1 mm, 2.6-μm reverse-phase liquid chromatography column, Kinetex^®^ C18 column (Phenomenex, CA, USA) at 40 °C. Furthermore, MS-grade solvents were used as the mobile phase in gradient elution mode: 0.1% aqueous formic acid as Solvent A and 0.1% formic acid in ACN as Solvent B. The flow was set to 0.22 mL/min with a gradient condition of Solvent B as 10% between 0 and 0.5 min, 10–50% between 0.5 and 21.5 min, 50–95% between 21.5 and 22.5 min, 95% between 22.5 and 25.5 min, 95–10% between 25.5 and 25.6 min, and 10% between 25.6 and 30 min.

### 2.5. Method Validation

Stock solutions of LOX, cis-LOX, and trans-Lox (40 mg/mL) were prepared in methanol to generate a calibration curve and linearity. The concentrations of LOX in plasma used to generate the calibration curve were 0.1, 0.2, 0.5, 1.0, 5.0, 10.0, 20.0, and 40.0 µg/mL. The concentrations of cis-LOX and trans-LOX used to generate their calibration curves were 0.2, 0.5, 1.0, 5.0, 10.0, 20.0, and 40.0 µg/mL. An amount of 10 µL of these samples was processed as mentioned above for LC-MS/MS analysis. Area peak ratios of analytes/IS versus concentration of samples were used to prepare the calibration curves. The calibration equation of LOX was *y* = 8 × 10^−7^*x* + 0.0002 (*R*² = 0.997). The calibration equations for cis-LOX and trans-LOX were *y* = 2 × 10^−6^*x* + 0.0002 (*R*² = 0.996) and *y* = 1 × 10^−6^*x* − 0.0002 (*R*² = 0.997), respectively.

To evaluate the accuracy and precision of LOX measurements, mouse plasma was spiked with a known concentration of LOX or QC samples at 0.2, 1.0, 10.0, 40.0 µg/mL (*n* = 5). Similarly, the accuracy and precision of cis-LOX and trans-LOX measurements were evaluated by spiking mouse plasma with a known concentration of either compound or QC samples at 0.5, 5.0, and 40.0 µg/mL (*n* = 5). Moreover, the accuracy and precision of intraday and interday were analyzed on the same day and five consecutive days at each concentration.

### 2.6. Statistical Analysis

The results are presented as the mean and the standard error of the mean. A Student’s unpaired *t*-test was applied in the statistical analyses of the obtained results using Graph Pad Prism. Results with a *p*-value ≤ 0.05, ≤ 0.01 and ≤0.001 were considered statistically significant.

## 3. Results 

### 3.1. Identification of Loxoprofen and Its Metabolites

To determine the plasma concentration of LOX and its metabolites via LC-MS/MS analysis, the produced product ions were checked and optimized for each compound. The ion intensities of LOX, cis-LOX, and trans-LOX were high in the negative mode of ionization; therefore, all the conditions for analysis used the negative mode for LC-MS/MS analysis. The MRM transitions chosen for LOX, cis-LOX, and trans-LOX were *m*/*z* 245.0 → 83.1, 247.1 → 202.2, and 247.1 → 203.1, respectively [15]. Representative MRM chromatograms of LOX, cis-LOX, trans-LOX, and the IS in mouse plasma are presented in Appendix A. LOX, cis-LOX, trans-LOX, and the IS were eluted at 8.5, 8.0, 8.2, and 8.9 min, respectively. No endogenous sources of interference were observed. LOX and its two metabolites were evaluated for linearity, precision, and accuracy. The calibration curves calculated within the range of 0.1–40.0 µg/mL for LOX and within the range of 0.2–40.0 µg/mL for cis-LOX and trans-LOX were linear. The precision (RSD %) range of LOX, cis-LOX, and trans-LOX was 1.8–12.9%, and the accuracy (RE %) range was less than 14.7% (Appendix A). Thus, the values were within the acceptable range and the method was accurate and precise.

### 3.2. Evaluation Model for Determination of LOX–Drug Interaction

To evaluate the CYP3A-induced LOX interaction, we prepared an experimental model that regulated CYP3A activity by administering an inducer (DEX) and an inhibitor (KTC) of CYP3A into mice. Briefly, DEX in corn oil was administered up to 3 consecutive days to induce CYP3A. Inhibition of CYP3A was induced by a single dose of KTC in 10% ethanol. Only vehicle groups were administered with their respective solvents without the addition of a CYP3A inducer or inhibitor. The induction and inhibition of CYP3A were confirmed with CYP assay using five different probe substrates (Appendix A).

DEX increased the metabolism of CYP3A substrate (midazolam) by approximately 10-fold when compared to the vehicle (VH) group. KTC significantly decreased the metabolism of CYP3A4 substrate (midazolam) when compared to the VH group (Appendix A). However, the metabolic activities of other CYP enzymes were unaffected between VH and treated groups. Thus, we validated our method of specifically regulating CYP3A activity via DEX and KTC.

### 3.3. Pharmacokinetic Analysis

The validated method was applied to determine the concentration of LOX, cis-LOX, and trans-LOX in mice pretreated with DEX or KTC. After pretreatment, LOX was orally administered to mice (20 mg/kg) after a 12-h fasting period. The plasma concentrations of LOX, cis-LOX, and trans-LOX were significantly decreased in the DEX pretreated group (Figure 1). In the KTC pretreated group, the plasma concentrations of cis-LOX, and trans-LOX were significantly increased and the plasma concentration of LOX was also increased but not significantly (Figure 1).

The PK parameters of LOX, cis-LOX, and trans-Lox in the VH- (corn oil) and DEX-treated groups are shown in Table 1. Although blood was collected for up to 240 min, LOX and its metabolites were not detected after 60 min. The *C*_max_, AUC_(0–60)_, and AUC_(0–∞)_ of all three compounds were significantly lowered in the DEX-pretreated group as opposed to the VH group. In the DEX-treated group, the *C*_max_ of LOX, cis-LOX, and trans-LOX was 2.5 ± 0.2, 1.1 ± 0.2, and 2.1 ± 0.2 µg/mL, respectively, and the AUC_(0–60)_ of the three compounds was 53.5 ± 6.1, 29.9 ± 4.4, and 67.6 ± 5.7 µg·min/mL, respectively. However, the *T*_max_ from all three compounds did not show any statistical difference when compared to the VH group. In contrast to the VH group, AUC_(0–∞)_ of LOX, cis-LOX, and trans-Lox in the DEX-treated group indicated a lower area under plasma concentration (56.2 ± 6.9, 31.5 ± 4.4, and 85.8 ± 5.0, respectively) over an extended time period. Moreover, the elimination *T*_1/2_ of LOX, cis-LOX, and trans-Lox in the VH group was 14.9 ± 0.6, 12.3 ± 0.3, and 18.2 ± 0.6 min, respectively, which is significantly different from the respective elimination *T*_1/2_ (12.0 ± 0.7, 13.9 ± 0.6, and 26.4 ± 1.6 min) generated by the DEX-pretreated group.

Furthermore, the PK parameters for LOX, cis-LOX, and trans-LOX in the VH and KTC groups are represented in Table 2. The *C*_max_ of cis-LOX, and trans-LOX in the VH group (1.2 ± 0.1, and 2.1 ± 0.2 µg/mL, respectively) were significantly lower than those in the KTC-treated group (1.6 ± 0.1, and 3.1 ± 0.3 µg/mL, respectively). However, the *T*_max_ of these three compounds did not show significant variation between the VH and KTC groups. In contrast to the *T*_max_, the elimination *T*_1/2_ of trans-LOX in the VH group (26.0 ± 0.5 min) was statistically different from that in the KTC-treated group (19.8 ± 0.7 min). Moreover, The AUC_(0–60)_ for, cis-LOX, and trans-LOX in the KTC-treated group was 49.0 ± 5.9, and 80.4 ± 9.6 µg·min/mL, respectively, which were higher than the respective values generated by the VH group. Altogether, the PK data generated by the DEX- and KTC-treated groups and their respective VH indicate that DEX and KTC significantly affected the PK of cis-LOX, and trans-LOX but PK of LOX was only affected by DEX, even though the formation of cis-LOX and trans-LOX was regulated by CR.

### 3.4. Metabolism and Metabolite Identification of LOX During DEX or KTC Treatment

The purpose of this study was to identify changes made by DEX and/or KTC on CR-mediated LOX metabolites (cis-LOX and trans-LOX) and also on CYP-mediated LOX metabolites (OH-LOX). During this PK study, OH-LOX could not be detected. Therefore, a full scan in Q Exactive Focus was used to identify all the metabolites present in plasma. A parallel reaction monitoring (PRM) mode was applied to confirm the metabolites through fragmentation patterns using collision induced dissociation (CID). Seven metabolites were confirmed after comparing the EICs of the test samples with blank plasma (Appendix A). LOX and all of its metabolites were detected in the negative ionization mode. To confirm the metabolites of LOX, their MS/MS fragmentation was checked (Appendix A). LOX (C_15_H_17_O_3_) was detected at a retention time of 18 min with only one major fragment ion 83.0492 (C_5_H_7_O). Trans-LOX (M1, C_15_H_19_O_3_) with an *m*/*z* ratio of 247.1339, eluted at 17.1 min with major fragment ions 233.1181 (C_14_H_17_O_3_, –CH_2_), 217.1230 (C_14_H_17_O_2_, –CH_2_O), 201.1279 (C_14_H_17_O), and 191.1071 (C_12_H_15_O_2_, –C_3_H_4_O). Cis-LOX (M2, C_15_H_19_O_3_), having the *m*/*z* ratio 247.1336, was detected at a retention time of 17.5 min. Its major fragments were 217.1230 (C_14_H_17_O_2_, –CH_2_O) and 191.1071 (C_12_H_15_O_2_, –C_3_H_4_O). M3 and M4 are OH-LOX (C_15_H_17_O_4_), having an *m*/*z* ratios of 261.1138 and 261.1133, and they were eluted at a retention time of 11.8 and 12.5 min, respectively. The MS/MS spectra of these metabolites were 99.0441 (C_5_H_7_O_2_, –C_10_H_10_O_2_) and 81.0335 (C_5_H_5_O, –C_10_H_12_O_3_) for M3 and only a single major product ion 99.0441 (C_5_H_7_O_2_) for M4. M5 is hydroxy trans-LOX (C_15_H_19_O_4_), having an *m*/*z* ratio of 263.1288, and it was detected at a retention time of 11 min. Its major fragment ions were 233.1181 (C_14_H_17_O_3_, –CH_2_O), 207.1022 (C_12_H_15_O_3_, –C_3_H_4_O), 133.0650 (C_9_H_9_O, –C_6_H_10_O_3_), and 99.0442 (C_5_H_7_O_2_, –C_10_H_12_O_2_). M6 was identified as a taurine conjugate (C_17_H_24_O_5_NS) whose *m/z* ratio was 354.1382, and it was detected at a retention time of 13.3 min. Its major fragments were 149.9859 (C_3_H_4_O_4_NS, –C_14_H_20_O), 124.0065 (C_2_H_6_O_3_NS, –C_15_H_18_O_2_), and 106.9798 (C_2_H_3_O_3_S, –C_15_H_21_O_2_N). M7 was a glucuronide conjugate (C_21_H_25_O_9_), having an *m/z* ratio 421.1514, and it eluted at a retention time of 14 min. Its major fragments were 245.1182 (C_15_H_17_O_3_), 193.0348 (C_6_H_9_O_7_), 175.0242 (C_6_H_7_O_6_), and 83.0492 (C_5_H_7_O). Parent compounds and their fragments detected during the study are represented in Table 3.

We identified the effects of CYP3A induction and inhibition on seven different known metabolites of LOX. The general characteristics of LOX, its metabolites, and their concentration in different groups are described in (Appendix A). In this study, we found that the concentration of LOX, trans-LOX (M1), and cis-LOX (M2) significantly decreased in the DEX-treated group (74.1 ± 6.3%, 80.1 ± 1.2%, and 61.9 ± 3.9%, respectively) and increased in the KTC-treated group (178.2 ± 8.3%, 158.9 ± 11.9%, and 173.1 ± 5.8%, respectively). Furthermore, the concentrations of M3, M4, and M5 significantly increased in the DEX-treated group (160.5 ± 4.1%, 440.4 ± 8.3%, and 286.3 ± 11.4%, respectively), and only the concentrations of M4 and M5 decreased in the KTC-treated group (93.6 ± 1.9% and 90.2 ± 2.7%, respectively). The taurine conjugate (M6) decreased in both the DEX- (65.3 ± 2.84%) and KTC-treated (91.2 ± 2.0%) groups. In contrast, the glucuronide conjugate increased in both the DEX- (174.4 ± 6.5%) and KTC-treated (275.7 ± 14.1%) groups. M6 and M7 are phase 2 metabolites, which indicates that CYPs are not the main enzyme involved in the formation of these metabolites. The concentration of LOX and its metabolites in the VH, DEX-treated, and KTC-treated groups is represented graphically in Figure 2. DEX and KTC are well-known CYP3A modulators, and in this study, we found that the pretreatment of DEX or KTC had a significant effect on the concentration of both CYP-mediated and CR-mediated metabolites.

DEX and KTC affected the concentration and the PK of CYP-mediated metabolites, which, in turn, influenced the concentration and the PK of cis- and trans-LOX. The metabolites detected in this study are summarized in Figure 3.

## 4. Discussion

LOX is a nonselective COX inhibitor that is administered for the management of pain and inflammation, and it is well tolerated by patients [2,24,25]. Although the pharmacokinetics and metabolism of LOX are well defined, its interaction(s) with CYP enzymatic pathways have not been fully characterized [3,6,7,8,9,10,13,15,26,27,28,29]. However, CYP3A regulating commercial drugs are widely in clinical use [16,17] and there may be the chance to use CYP3A-regulating drugs and LOX together. Although CYP3A does not generate the LOX active metabolite, it may be possible to influence its generation by regulating CYP3A activity. To that end, we evaluated whether the rate of LOX active metabolite formation is influenced by CYP3A activity. We also investigated the effects of DEX and KTC, which are widely used CYP3A modulators, on the metabolism and PK of LOX, cis-LOX, and trans-LOX.

DEX is a steroidal drug used in the treatment of many conditions such as skin diseases, allergies, rheumatic disorder, asthma, and certain autoimmune diseases [30,31]. It is also effective in treating various cancers such as central nervous system tumors, brain metastases, advanced melanoma, leukemia, lymphoma, and multiple myeloma. Additionally, DEX is effective at combating the side effects of chemotherapy [32,33]. DEX is metabolized by CYP3A4 and CYP17A, but the CYP17A metabolic pathway plays no major role in its in vivo metabolism [34,35,36]. DEX has been used as a CYP3A4 inducer in clinical studies. Interestingly, rifampicin, rifabutin, phenytoin, phenobarbital, primidone, carbamazepine, etc., are strong CYP3A4 inducers [16]. It has also been reported that a single dose (50 mg/kg) of DEX can induce the activity of CYP3A. However, persistent administration of DEX could make stable induction of CYP3A activity, which was independent of inducer administration [37,38]. Moreover, chronic DEX administration likely leads to the autoinduction of CYP3A, which has a direct impact on the PK parameters of DEX [38]. Nevertheless, patients may suffer from an increase in substrate concentration after they stop using the CYP3A inducer if they take both the substrate (LOX) and the inducer (DEX) of CYP3A [37].

KTC is a broad-spectrum antifungal drug used in the treatment of many fungal infections such as blastomycosis, coccidioidomycosis, histoplasmosis, etc. [39,40]. KTC is a known reversible inhibitor of CYP3A4 [16] and displays hepatotoxicity through immune-allergic mechanisms, which limits its therapeutic use [41,42]. Usually, CYP3A4 inhibitors are divided into reversible inhibitors (such as KTC, itraconazole, terfenadine, astemizole, and quinidine) [43] and irreversible inhibitors (such as gestodene and levonorgestrel) [44]. Many studies also reported the biotransformation of KTC through oxidation, O-dealkylation, hydroxylation, and FMO (via the UGT1A4 metabolic pathway) [42,45]. Furthermore, one study revealed that KTC showed dose-dependent kinetics, indicating that its metabolism occurs in the liver [46]. A recent study also revealed that KTC metabolism is similar in humans and mice, which may help to resolve the issue regarding drug metabolism and toxicology [42]. Nevertheless, a sudden increase or decrease in the *C*_max_ of drugs and toxic metabolites could result from either the induction or inhibition of metabolizing enzymes [47].

CYP3A4 is the major CYP involved in LOX metabolism, but it does not form its active metabolite [15]. The active metabolite of LOX (trans-LOX) is formed via CR pathway [1,5]. Similarly to previous studies, we also found that DEX and KTC were strong CYP3A activity modulators [36,48,49]. In addition, we determined that the *C*_max_, AUC_(0–60)_, and AUC_(0__–__∞)_ of LOX, cis-LOX, and trans-LOX were significantly decreased in DEX-treated mice. However, the *C*_max_, AUC_(0–60)_, and AUC_(0__–__∞)_ of cis-LOX, and trans-LOX are significantly increased in KTC-treated mice. Furthermore, DEX increased the concentration of the OH-LOX metabolite and decreased the concentration of the active metabolite. This may lead to a decrease in the activity of trans-LOX during the subsequent administration of LOX and DEX. We also observed that changes in CYP pathway activity can modulate CR pathway activity regarding LOX, even though CYP3A4 does not participate in LOX active metabolite formation.

These results indicate that the co-administration of LOX and DEX may lead to a decrease in the pharmacological activity of LOX by decreasing the concentration of the active metabolite. We also observed that the concentration of active metabolites catalyzed by CR increased and that the concentration of inactive metabolites decreased during the co-administration of LOX and the CYP3A inhibitor, KTC. Therefore, if LOX and KTC are administered together, the pharmacological activity of LOX may be enhanced by increasing the active metabolite. Subsequently, this may also increase toxicity. In a previous study, LOX was associated with an increase in small bowel mucosal injury, erosive gastritis, gastroduodenal ulcers, etc., during concomitant use of a proton pump inhibitor such as lansoprazole [1,50,51]. Since lansoprazole is a known inhibitor of CYP3A and CYP2C19 [52,53], these side effects may have been caused by changes in the blood levels of LOX and its active metabolite caused by the concomitant use of LOX and lansoprazole. It has been reported that LOX slightly inhibited the metabolism of tacrolimus by interacting with CYP3A in human liver microsomes [54]. Similarly, tizanidine is typically used to manage muscle spasticity and pain [55]. However, it has been reported that combination therapy of tizanidine and LOX for neck pain increased the risk of irreversible symptomatic bradycardia via CYP inhibition [56].

These previous studies clearly show that LOX can interact with drugs that modulate CYP activity even though its active metabolite is not formed by CYP enzymes. Considering the pharmacokinetics and metabolism of LOX and its metabolites, the use of LOX with CYP modulators (e.g., DEX and KTC) may result in the decreased pharmacological action of LOX or may cause from minor toxicity to major toxicity by interacting with CYP3A.

## 5. Conclusions

The pharmacokinetic parameters of LOX and its active metabolite were significantly altered when LOX was co-administered with CYP3A activity modulators. In clinical practice, LOX is administered concurrently with many other drugs. Therefore, more studies are needed to assess the possible interactions of LOX with CYP enzymes. In this study, pharmacodynamic interactions were not evaluated; however, they will be evaluated in future studies.

## Figures and Tables

**Figure 1 pharmaceutics-11-00479-f001:**
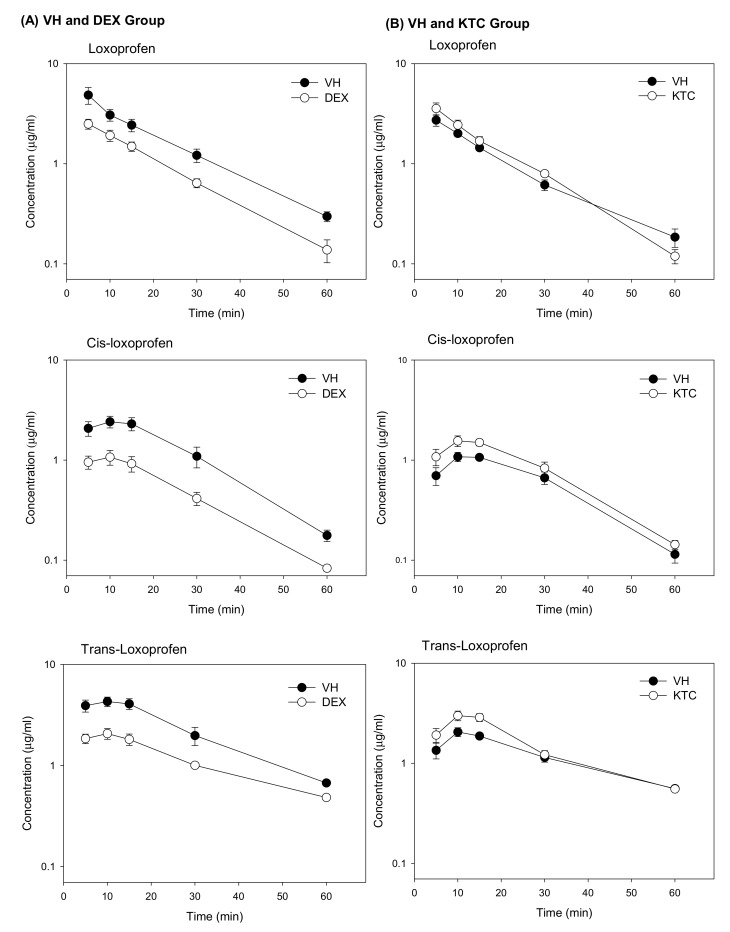
Mean plasma concentration versus time profiles of LOX, cis-LOX, and trans-LOX in either the presence of a CYP3A4 inducer (DEX) or inhibitor (KTC) with their respective vehicle. (**A**) Mean plasma concentration versus time profiles after *i.p.* administration of either VH (corn oil) or DEX (40 mg/kg) for 3 consecutive days. The plasma concentrations of all the compounds in the VH and DEX groups showed significant decrement up to 60 min in the DEX-treated group as compared to its VH group. The bars represent standard error (SE) (*n* = 3). (**B**) Mean plasma concentration versus time profiles after a single dose *i.p.* administration of either VH (10% ethanol) or KTC (60 mg/kg). In the KTC group, LOX, cis-LOX, and trans-LOX showed increased mean plasma concentrations as opposed to the VH group. The bars indicate standard error (SE) (*n* = 3).

**Figure 2 pharmaceutics-11-00479-f002:**
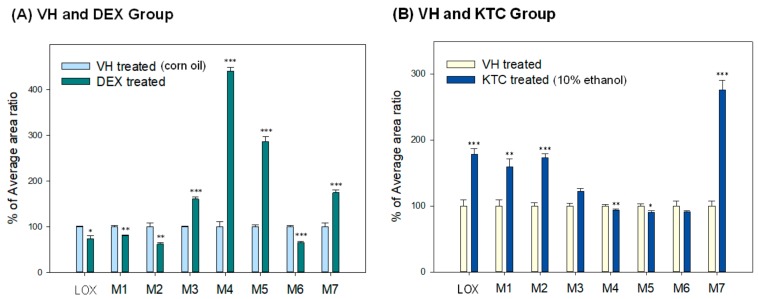
Comparison of loxoprofen (LOX) and its metabolites in male ICR mice treated with VH, DEX, or KTC. (**A**) The relative concentration of LOX and its metabolites after DEX administration (*i.p.* 40 mg/kg for consecutive 3 days, *n* = 3) compared to VH (*i.p.* corn oil for consecutive 3 days, *n* = 3). (**B**) The relative concentration of LOX and its metabolites after KTC administration (single dose *i.p.* 60 mg/kg, *n* = 3) compared to VH (10% ethanol, *n* = 3).

**Figure 3 pharmaceutics-11-00479-f003:**
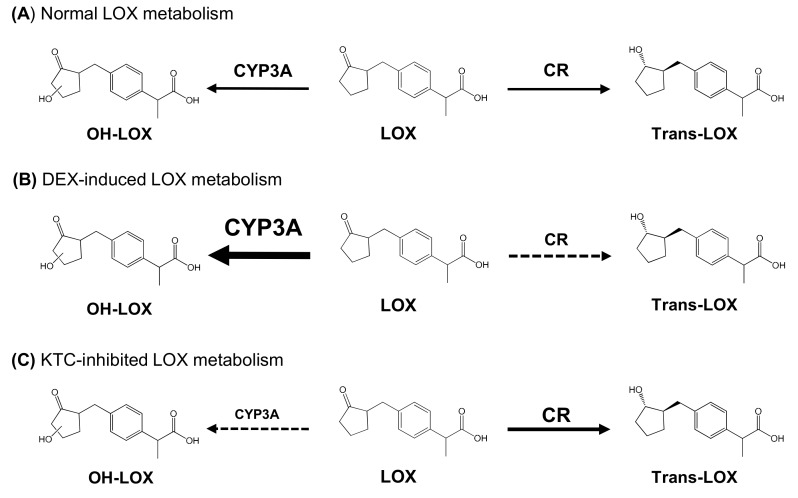
Metabolic pathway of loxoprofen and its metabolites.

**Table 1 pharmaceutics-11-00479-t001:** Pharmacokinetic parameters of loxoprofen (LOX), cis-LOX, and trans-LOX in VH- (corn oil) and dexamethasone (DEX)-treated groups.

Analytes	Parameters	VH (Corn Oil)	DEX
LOX	*C*_max_ (µg/mL)	4.8 ± 0.9	2.5 ± 0.2 **
	*T*_max_ (min)	5.0 ± 0.0	5.0 ± 0.0
	AUC_(0–60)_ (µg·min/mL)	95.7 ± 14.5	53.5 ± 6.1 **
	*T*_1/2_ (min)	14.9 ± 0.6	12.0 ± 0.7 *
	AUC_(0–∞)_ (µg·min/mL)	102.2 ± 15.0	56.2 ± 6.9 ***
cis-LOX	*C*_max_ (µg/mL)	2.4 ± 0.3	1.1 ± 0.2 **
	*T*_max_ (min)	10.4 ± 0.7	10.0 ± 1.0
	AUC_(0–60)_ (µg·min/mL)	72.7 ± 12.4	29.9 ± 4.4 **
	*T*_1/2_ (min)	12.3 ± 0.3	13.9 ± 0.6 *
	AUC_(0–∞)_ (µg·min/mL)	75.8 ± 12.8	31.5 ± 4.4 **
trans-LOX	*C*_max_ (µg/mL)	4.4 ± 0.5	2.1 ± 0.2 **
	*T*_max_ (min)	9.1 ± 1.2	10.4 ± 0.9
	AUC_(0–60)_ (µg·min/mL)	137.4 ± 19.0	67.6 ± 5.7 **
	*T*_1/2_ (min)	18.2 ± 0.6	26.4 ± 1.6 **
	AUC(_0–∞_) (µg·min/mL)	154.8 ± 19.2	85.8 ± 5.0 **

All data are expressed as the mean ± standard error (SE) (*n* = 3). *C*_max_: maximum plasma concentration; AUC_(0–60)_: area under the plasma concentration-time curve (AUC) from 0 to 60 min; *T*_max_: time to reach maximum plasma concentration; *T*_1/2_: elimination half-life; AUC_(0–∞)_: area under the plasma concentration-time curve from 0 to infinite time. * *p* ≤ 0.05, ** *p* ≤ 0.01 and *** *p* ≤ 0.001.

**Table 2 pharmaceutics-11-00479-t002:** Pharmacokinetic parameters of LOX and its metabolites in 10% ethanol (VH) and KTC-treated groups.

Analytes	Parameters	VH (10% Ethanol)	KTC
LOX	*C*_max_ (µg/mL)	2.7 ± 0.3	3.5 ± 0.5
	*T*_max_ (min)	5.0 ± 0.0	5.0 ± 0.0
	AUC_(0–60)_ (µg·min/mL)	54.7 ± 4.6	66.6 ± 6.6
	*T*_1/2_ (min)	14.6 ± 1.1	12.2 ± 0.9
	AUC_(0–∞)_ (µg·min/mL)	59.1 ± 5.6	68.9 ± 6.5
cis-LOX	*C*_max_ (µg/mL)	1.2 ± 0.1	1.6 ± 0.1 *
	*T*_max_ (min)	13.3 ± 0.8	11.6 ± 0.8
	AUC_(0–60)_ (µg·min/mL)	36.0 ± 3.9	49.0 ± 5.9 *
	*T*_1/2_ (min)	13.3 ± 0.9	13.4 ± 0.6
	AUC_(0–∞)_ (µg·min/mL)	38.4 ± 4.4	51.9 ± 6.1 *
trans-LOX	*C*_max_ (µg/mL)	2.1 ± 0.2	3.1 ± 0.3 **
	*T*_max_ (min)	11.6 ± 0.8	11.7 ± 0.8
	AUC_(0–60)_ (µg·min/mL)	69.9 ± 6.0	80.4 ± 9.6 *
	*T*_1/2_ (min)	26.0 ± 0.5	19.8 ± 0.7 **
	AUC_(0–∞)_ (µg·min/mL)	90.9 ± 7.1	94.7 ± 11.0

All data are expressed as the mean ± standard error (SE) (*n* = 3). *C*_max_: maximum plasma concentration; AUC_(0_*_–_*_60)_: area under the plasma concentration-time curve (AUC) from 0 to 60 min; *T*_max_: time to reach maximum plasma concentration; *T*_1/2_: elimination half-life; AUC_(0_*_–_*_∞)_: area under the plasma concentration-time curve from 0 to infinite time. * *p* ≤ 0.05, ** *p* ≤ 0.01 and *** *p* ≤ 0.001.

**Table 3 pharmaceutics-11-00479-t003:** Identified metabolites of LOX in mouse plasma using HRMS.

Compounds	Parent Ions (*m*/*z*)	Elemental Composition	Error (ppm)	Product Ions (*m*/*z*)	Description
Lox	245.1179	C_15_H_17_O_3_	0.4	83.0492	LOX
M1	247.1339	C_15_H_19_O_3_	2.0	233.1181, 217.1230, 201.1279, 191.1071	Trans-LOX
M2	247.1336	C_15_H_19_O_3_	0.5	217.1230, 191.1071	Cis-LOX
M3	261.1138	C_15_H_17_O_4_	4.2	99.0441, 81.0335	OH-LOX
M4	261.1133	C_15_H_17_O_4_	2.3	99.0441	OH-LOX
M5	263.1288	C_15_H_19_O_4_	1.9	233.1181, 207.1022, 133.0650, 99.0442	OH-trans-LOX
M6	354.1382	C_17_H_24_O_5_NS	2.0	149.9859, 124.0065, 79.9563	Taurine conjugate
M7	421.1514	C_21_H_25_O_9_	3.6	245.1182, 193.0348, 175.0242, 83.0492	Glucuronide conjugate

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
