# Peer review of "Assessing Drug Interaction and Pharmacokinetics of Loxoprofen in Mice Treated with CYP3A Modulators"

_pharmaceutics, 2019, doi:10.3390/pharmaceutics11090479_

Round 1

Reviewer 1 Report

In this manuscript, authors investigated the metabolism of LOX after the induction or inhibition of activities of CYP3A, respectively. In general, the experiments were reasonably designed and well performed. However, some concerns were listed as below:

Major concerns:

In the section 2.3. (CYP activities in the mouse liver), for the inhibition experiment, the liver was excised 24 hours after the last treatment of KTC. The KTC has been excreted from liver and the CYP3A activity may be already recovered. The data in Supplemental Figure 2B also indicated that CYP3A4 was not dramatically inhibited (less than 20%). Early time point may be more ideal for this inhibition study. In figure 1, the plots only showed the time points up to 60 min. However, in the table, AUC(0-240) (μg·min/ml) were presented. Please clarify how these parameters calculated. In Figure 2, authors assumed each metabolite 100% in VH treated group, which may confused the readers. In Fig. 2B, it looks like that LOX is less metabolized and more metabolites generated. If authors assume all the metabolites as 100% in VH groups, KTC group, and DEX group individually, the percentage of each metabolite were compared, which may make more sense.

Minor concerns:

Supplemental Figure 2B did not include the data about CYP2B and 2C. On page 5, lines 212 and 214, and Page 7, line 244 what does “Error! Reference source not found”? (Please go through the manuscript, multiple places were described like this) On page 11, Figure 1 should be Figure 3.

Author Response

Q) In the section 2.3. (CYP activities in the mouse liver), for the inhibition experiment, the liver was excised 24 hours after the last treatment of KTC. The KTC has been excreted from liver and the CYP3A activity may be already recovered. The data in Supplemental Figure 2B also indicated that CYP3A4 was not dramatically inhibited (less than 20%). Early time point may be more ideal for this inhibition study.

>> Thank you so much for your kind suggestion. The experimental method used in this experiment was conducted according to the mouse PK method developed in our previous report (Jo et al., 2018). In a further study, we will design an experiment that reflects the reviewer's opinion.

Jo JJ, Jo JH, Kim S, Lee JM, Lee S. Development of a simultaneous LC-MS/MS method to predict in vivo drug-drug interaction in mice. Arch Pharm Res. 2018 41(4):450-458.

Q) In figure 1, the plots only showed the time points up to 60 min. However, in the table, AUC(0-240) (μg·min/ml) were presented. Please clarify how these parameters calculated.

>> We actually collected blood upto 240 min for PK study but the compounds were not detected after 60 min so in the table there should be AUC (0-60) (μg·min/ml). We have changed the label in table and text.

Q) In Figure 2, authors assumed each metabolite 100% in VH treated group, which may confuse the readers. In Fig. 2B, it looks like that LOX is less metabolized and more metabolites generated. If authors assume all the metabolites as 100% in VH groups, KTC group, and DEX group individually, the percentage of each metabolite were compared, which may make more sense.

>> Thank you for your comment. We agree to reviewer’s comment. VH group in DEX-treated group was treated with corn oil, whereas VH group in KCT-treated group was administered once with 10% ethanol. To distinguish between the VH-groups in each group, we have added an additional label to the figure.

Minor concerns:

Q) Supplemental Figure 2B did not include the data about CYP2B and 2C.

>> Regarding CYP2B and 2C, we did not obtain data during CYP inhibition assay. So, we could not include in manuscript. We deleted the description for CYP2B and 2C in Method section.

Q) On page 5, lines 212 and 214, and Page 7, line 244 what does “Error! Reference source not found”? (Please go through the manuscript, multiple places were described like this) On page 11, Figure 1 should be Figure 3. 

>> Thank you for the kind indication, we corrected the errors in references. And figure 1 from page 11 are made figure 3.

Reviewer 2 Report

The authors demonstrated the pharmacokinetic interaction between loxoprofen (LOX) and CYP3A modulators in mice.  Overall, the manuscript is well written, the aim of the study is sound, and the interpretation of the results are reasonable. The conclusion is extremely interest on the consideration of appropriate use of LOX.  Therefore, the reviewer recommend that the article is published in the journal of "Pharmaceutics," following minor revisions.

Authors should reconfirm whether the word initials in the title should be upper or lower case. For example, isn't the initial letter (m) of the modulator a capital letter? Dexamethasone (DEX; 40 mg/kg) was administered intraperitoneally (ip) to mice for 3 consecutive days, and then LOX was administered orally after fasting. In addition,  ketoconazole was (KTC; 60 mg/kg) administered ip to mice, and after 3 min, LOX was given orally.  The authors should provide a reference based on the processing conditions of DEX and KTC. Authors described that trans-LOX was active metabolite. The authors should add to which of trans-LOX and LOX has higher pharmacological activity in this text. Authors should add whether the major metabolic pathway of LOX is through carbonyl reductase or CYP3A.

Author Response

Q) Authors should reconfirm whether the word initials in the title should be upper or lower case. For example, isn't the initial letter (m) of the modulator a capital letter?

>> Following reviewer’s comment, all the initials in the title are made upper case.

Q) Dexamethasone (DEX; 40 mg/kg) was administered intraperitoneally (ip) to mice for 3 consecutive days, and then LOX was administered orally after fasting. In addition, ketoconazole was (KTC; 60 mg/kg) administered ip to mice, and after 3 min, LOX was given orally.  The authors should provide a reference based on the processing conditions of DEX and KTC.

>> Thank you for your comment. We added references regarding the processing of DEX and KTC (In line 88 and 91). The methods from references are slightly modified and used in our study.

Q) Authors described that trans-LOX was active metabolite. The authors should add to which of trans-LOX and LOX has higher pharmacological activity in this text.

>> LOX is a prodrug which is metabolized by carbonyl reductase and converted into its active metabolite trans-LOX. LOX is pharmacologically inactive drug unless it is metabolized to trans-LOX. It is additionally marked in the Introduction section.

Q) Authors should add whether the major metabolic pathway of LOX is through carbonyl reductase or CYP3A. 

>> The major metabolic pathway of LOX is carbonyl reductase mediated pathway because its active metabolite (trans-LOX) is formed by carbonyl reductase. On the other hand, its inactive metabolite hydroxy-LOX is formed by CYP3A enzyme which is the minor metabolic pathway. It is additionally marked in the Introduction section.

Reviewer 3 Report

The authors examined the pharmacokinetics and drug interactions of loxoprofen in young male mice. The experimental design appears rather straightforward, but there are several major concerns regarding the methodology and statistical analyses.

For the PK study, blood samples were collected from the tail over time. How much volume of blood was collected? It is questionable whether the volume collected from a 6-week old mouse would be enough for the assays performed?

Some studies were a within-design (like the PK study), whereas other between subjects. The distinction is not clear. It is even more confusion since the authors list an unpaired-t-test as the only statistics performed. This is not appropriate. 

The total number of mice were not listed.

The method of euthanasia is not listed.

There is not citation or rationale for pre-treatment for the CYP inhibitors. Why was the the time/dose chosen?    

Line 212, 214 and 307 "Error!" is noted for the references.

Author Response

Q) For the PK study, blood samples were collected from the tail over time. How much volume of blood was collected? It is questionable whether the volume collected from a 6-week old mouse would be enough for the assays performed?

>> Thank you for your comments. For PK study we collected 20~30 µl of blood from tail at each time interval. From the collected blood 10 µl of plasma was extracted and used for the study. The plasma samples were enough for us to perform the study. The experimental method used in this experiment was conducted according to the mouse PK method developed in our previous report (Jo et al., 2018).

Q) Some studies were a within-design (like the PK study), whereas other between subjects. The distinction is not clear. It is even more confusion since the authors list an unpaired-t-test as the only statistics performed. This is not appropriate. 

>> In this study, Student’s paired t-test was used to verify statistical significance between the two groups. Statistical significance was verified between VH and DEX treatment groups and between VH and KCT treatment groups, respectively.

Q) The total number of mice were not listed.

>> Total number of mice used in the study were: 36 mice

Q) The method of euthanasia is not listed.

>> The mice used for this study were sacrificed by cervical dislocation after last blood collection (line 95).

Q) There is not citation or rationale for pre-treatment for the CYP inhibitors. Why was the the time/dose chosen?  

 >> Thank you for your comment. We added references regarding the processing of DEX and      KTC. The methods from references are slightly modified and used in our study.

Q) Line 212, 214 and 307 "Error!" is noted for the references.

>> Thank you for the kind indication, we corrected the errors in references.

Reviewer 4 Report

This study aimed that concomitant use of loxoprofen with CYP3A modulators may lead to drug-drug interactions and result in minor to severe toxicity even though there is no direct change in the metabolic pathway that forms the loxoprofen active metabolite. The manuscript is well presented and written and the finding look reliable. Hovewer some minor changes and clarifications along the manuscript are needed.

Comments

2.1. Materials

Please for all the chemical used in this study include the chemical name, chemical formula, CAS number as well the purity and solubility rate.

Please justify why only male mice were used in this study.

 3.3. Pharmacokinetic analysis

Lines 213-214. Please check this sentence and clarify “The PK parameters of LOX, cis-LOX, and trans-Lox in the VH- (corn oil) and DEX-treated groups are shown in Error! Reference source not found.

Lines 243-244- Please check this sentence and clarify “Furthermore, the PK parameters for LOX, cis-LOX, and trans-LOX in the VH and KTC groups are represented in Error! Reference source not found.

Please change T1/2 to read elimination T1/2.

Author Response

Q) 2.1. Materials: Please for all the chemical used in this study include the chemical name, chemical formula, CAS number as well the purity and solubility rate.

>> Following reviewer’s comments, we added the information of loxoprofen and trans-loxoprofen as main compounds, cis-loxoprofen doesn’t have CAS number.

Q) Please justify why only male mice were used in this study.

>> Male mice are used for general toxicity evaluation and PK experiments. Female mice may cause changes in physiological function due to hormonal changes in each individual, so use male individuals except for experiments with special purpose. 3.3. Pharmacokinetic analysis

Q) Lines 213-214. Please check this sentence and clarify “The PK parameters of LOX, cis-LOX, and trans-Lox in the VH- (corn oil) and DEX-treated groups are shown in Error! Reference source not found.”

>> Thank you for the kind indication, we corrected the errors in references.

Q) Lines 243-244- Please check this sentence and clarify “Furthermore, the PK parameters for LOX, cis-LOX, and trans-LOX in the VH and KTC groups are represented in Error! Reference source not found.”

>> We corrected the errors in references.

Q) Please change T1/2 to read elimination T1/2.

>> Thank You. We have changed T1/2 to elimination T1/2.

Round 2

Reviewer 3 Report

The authors have not addressed all my concerns, in my opinion the statistics and degrees of freedom, and sample sizes are unclear. I suggest the authors consult a statistician. 

Author Response

Question) The authors have not addressed all my concerns, in my opinion the statistics and degrees of freedom, and sample sizes are unclear. I suggest the authors consult a statistician.

(Previous question) Some studies were a within-design (like the PK study), whereas other between subjects. The distinction is not clear. It is even more confusion since the authors list an unpaired-t-test as the only statistics performed. This is not appropriate. 

 [Answer] first, we are sorry for not responding properly to the reviewer's requirements. We used paired-t-test for statistical processing in our paper, because the statistical method was selected by referring to other papers with the same type of results. The results agree with reviewers who pointed out that unpaired-t-test is a better statistical method than paired-t-test for this study. The research results were again statistically processed by unpaired t-test, and the relevant research contents were corrected. Although the significant of LOX-related pharmacokinetic parameters has changed in the Ketoconazole-treated group, the overall orientation of the paper remains unchanged. In our study we used triplicate samples for each group. So, the sample size in each group was n=3 and the degree of freedom of each group is 2 (n-1). We have added sample size in manuscript. Total number of mice used in our study were 36 mice; 12 mice for CYP inhibition assay, 12 mice for Pharmacokinetic study and 12 mice for metabolite identification. We apologize again for not understanding the reviewer's suggestion and thank you for pointing it out.

Round 3

Reviewer 3 Report

The authors’ explanation of the data analysis is sufficient. Thank you for including the sample size.